# Ultra-Broadband Perfect Absorber based on Titanium Nanoarrays for Harvesting Solar Energy

**DOI:** 10.3390/nano13010091

**Published:** 2022-12-24

**Authors:** Didi Song, Kaihua Zhang, Mengdan Qian, Yufang Liu, Xiaohu Wu, Kun Yu

**Affiliations:** 1Henan Key Laboratory of Infrared Materials & Spectrum Measures and Applications, School of Physics, Henan Normal University, Xinxiang 453007, China; 2Shandong Institute of Advanced Technology, Jinan 250100, China

**Keywords:** metamaterial, perfect absorber, ultra-broadband absorption, polarization independence

## Abstract

Solar energy is a clean and renewable energy source and solves today’s energy and climate emergency. Near-perfect broadband solar absorbers can offer necessary technical assistance to follow this route and develop an effective solar energy-harvesting system. In this work, the metamaterial perfect absorber operating in the ultraviolet to the near-infrared spectral range was designed, consisting of a periodically aligned titanium (Ti) nanoarray coupled to an optical cavity. Through numerical simulations, the average absorption efficiency of the optimal parameter absorber can reach up to 99.84% in the 200–3000 nm broadband range. We show that the Ti pyramid’s localized surface plasmon resonances, the intrinsic loss of the Ti material, and the coupling of resonance modes between two neighboring pyramids are highly responsible for this broadband perfect absorption effect. Additionally, we demonstrate that the absorber exhibits some excellent features desirable for the practical absorption and harvesting of solar energy, such as precision tolerance, polarization independence, and large angular acceptance.

## 1. Introduction

Solar energy is an emerging renewable energy source. To efficiently collect and utilize solar energy, it is urgent to develop a wideband and efficient solar energy absorber. A perfect solar absorber should have a near-unity absorption bandwidth close to the solar spectrum [1,2,3]. Metamaterials, as a new type of artificial electromagnetic medium, are composed of sub-wavelength-unit structure supporting the periodic, quasi-periodic, or random arrangement mode, and they not only inherit a part of characteristics of natural materials, but also obtain electromagnetic properties that surpass natural materials [4,5,6,7,8]. Since Landy first proposed the metamaterial perfect absorber with a 100% absorption at a specific frequency in 2008 [9], the designable electromagnetic modulation properties of metamaterials have stimulated researchers’ interest in various fields. Studies have shown that the energy generated by solar radiation is mainly concentrated in the ultraviolet (UV) to the near-infrared (NIR) range, which can be used in fields such as solar energy harvesting [10,11,12], thermophotovoltaics [13,14], and photoelectric detection [15,16]. Consequently, it is essential to research the perfect absorber of metamaterials in the UV to the NIR band for the development and utilization of solar energy. 

The representative designed structure of the metamaterial absorber consists of a sandwich “metal−insulator−metal” (MIM) layer, in which metallic resonators are periodically arranged on top of an engineering double-layer substrate (dielectric/metal) [17,18,19,20]. Perfect absorption can be achieved in the situation where the impedance of the system matches the ambient impedance. Until now, great efforts have been made to develop highly efficient broadband absorbers [21,22,23,24,25,26]. For example, Hao et al. [27] designed a non-tapered symmetrical Au−Si multilayered conical frustum structure, which can realize high absorption in the wavelength range of 480−1480 nm. Mou et al. [28] fabricated a centimeter-sized metamaterial sample of a metal−insulator−metal structure, in which Ge_2_Sb_2_Te_5_ is used as the insulation interval of the MIM structure, with an absorption efficiency of 80% in a broadband (480–1020 nm). Yu et al. [11] demonstrated that the surface plasmon resonance is generated by the interaction of various surface microstructures, resulting in a wide electromagnetic absorption spectrum of the absorber, with an average absorption efficiency which can reach 93.17% at 100–2000 nm. However, these broadband absorbers still have one or more disadvantages, such as complex nanostructure, expensive manufacturing cost, and the working bandwidth that does not fully cover the UV and IR spectral ranges.

Moreover, precious metals such as gold (Au) or silver (Ag) are extensively exploited materials for previously reported metamaterial absorbers. However, they have deficiencies such as weak thermal stability, a narrow absorption spectrum, and expensive costs. In contrast, Ti is an excellent refractory material with a melting point of 1668 ℃, high thermal durability, and superior corrosion resistance and has been proven to replace precious metals in perfect absorption applications [11,29,30]. Ti is also relatively inexpensive compared to precious metals. Particularly, Ti has high intrinsic loss and displays excellent plasmonic resonance in the visible (VIS) to the NIR range, favorable to the enhancement of the plasmonic absorption in this spectral range [18,20]. These features can satisfy the demands of solar thermal photovoltaic systems. 

In this paper, we designed a metamaterial broadband perfect absorber made of Ti operating in the UV and NIR regions. By designing a periodic Ti pyramid in the topmost metal layer, the absorption bandwidth can reach 2800 nm with a minimal absorptivity over 99.4%, and the maximal absorptivity reaches 99.99% from the UV to NIR range. We systemically studied the influence of various geometrical parameters, polarization states, and incident angles on the absorption spectrum, which helps us to have a comprehensive understanding of the absorption mechanism of the designed absorber. It was demonstrated that the designed absorber exhibits polarization independence and large angular acceptance across the entire spectrum from 200 to 3000 nm at normal incidence.

## 2. Materials and Methods

Figure 1a shows the schematic geometry of the designed absorber and the unit cell. The absorber consists of a Ti pyramid array and a silicon dioxide (SiO_2_) dielectric film on a Ti substrate. The height of the Ti pyramid with the same periodicity P is h_1_ and arranged in both the x- and y-directions. The heights of the SiO_2_ dielectric layer and the Ti substrate layer are h_2_ and h_3_, respectively. The material refractive indexes of Ti and SiO_2_ can be obtained from the Palik’s database [31,32]. Herein, 3D ultra-broadband simulations of the designed absorber were carried out by using commercial software, COMSOL Multiphysics. Using a Floquet port at the top and bottom of the structure, we designed a periodic structure with boundary conditions. Furthermore, the matching perfectly matched layers are intended to banish surface scattering. As shown in Figure 1(a), a transverse-electric (TE) polarized plane wave with a polarization parallel to the x-direction is incident onto the absorber along the z-direction with wavelengths ranging from 200 to 3000 nm. 

The absorption efficiency can be expressed by the equation:(1)$$\begin{array}{c} \;A\left(\lambda \right)=1-R\left(\lambda \right)-T\left(\lambda \right), \end{array}$$
where ***A*** is the light absorption, ***R*** is the light reflection, and ***T*** is the light transmission. Since the thickness of the Ti substrate is much greater than its skin depth, the transmitted light is prohibited so that the absorption can be represented as ***A***(***λ***) = **1** − ***R***(***λ***). When the reflected light of the absorber is minimized, a near-unity absorption can be obtained. Our proposed structure can be readily fabricated using traditional nanotechnologies. Figure 1b depicts the simulated absorption spectra of the designed absorber at the optimal geometric parameters: h_1_ = 800 nm, h_2_ = 30 nm, h_3_ = 110 nm, and P = 100 nm. The designed absorber exhibits the absorptivity over 99.4% in the whole spectrum (200–3000 nm), with an average absorption efficiency reaching 99.84%. More importantly, the absorption bandwidth with an absorption efficiency of greater than 99% reaches 2800 nm, indicating that the ultra-wideband perfect absorption is achieved.

## 3. Results

### 3.1. Perfect Absorption Performance

For the purpose of assessing the potential application for solar energy collection, we had a thorough study of the solar absorption behavior of metamaterial perfect absorbers. By calculating the absorption of the designed absorber under the air mass (AM) 1.5 illumination and comparing it with the AM 1.5 spectrum, we evaluated the solar absorption performance of the absorber. Obviously, the energies radiated by the solar are mainly focused in the ranges of 280–2500 nm and 3000–4000 nm, especially in the visible to near-infrared regions. To improve the solar light absorption, we must enhance the absorption performance in the mentioned range. It can be seen in Figure 2a that under simulated solar irradiation, the spectral curves of the absorber and the standard solar source almost coincide, while the spectral curve of unabsorbed solar energy is close to 0. According to the calculation, the metamaterial perfect absorber can absorb more than 99.93% of the solar energy, showing perfect ultra-broadband spectral absorption within the spectrum of 280–4000 nm. We also performed its planar counterpart simulations, with the same geometric parameters and materials, confirming the structural advantages of the designed absorber. Figure 2b compares the absorbed energies of the pyramidal structure and its planar counterpart under AM 1.5 illumination. Over the whole examined wavelength area (280–4000 nm), the designed structure’s absorption is markedly superior to that of the planar structure, demonstrating that addition of tapered structural features leads to a significant enhancement in the absorption performance.

### 3.2. Absorption Mechanism

The broadband high absorption characteristics of metamaterials to comprise perfect solar absorbers were analyzed combined with the impedance matching theory. When the incident light travels vertically to the surface of the absorber structure, the effective impedance Z of the absorber is expressed as [33,34]: (2)$$\begin{array}{c} \;Z=\sqrt{\frac{{\left(1+S11\right)}^{2}-S21^{2}}{{\left(1-S11\right)}^{2}-S21^{2}}\;},\# \end{array}$$
where *S*11/*S*21 is the reflection/transmission coefficient. Since the substrate thickness of the designed absorber is greater than the skin-seeking depth, Equation (2) can be shortened to the following:(3)$$\begin{array}{c} \;Z=\sqrt{\frac{{\left(1+S11\right)}^{2}}{{\left(1-S11\right)}^{2}}}. \end{array}$$

Evidently, near-unity absorption can be achieved, when the value of the effective impedance *Z* includes the real part (near 1) and the imaginary part (near zero). Figure 3 shows the calculated result from Equation (3), demonstrating the effective impedance Z matching that in the free space. Therefore, our metamaterial solar absorber has a better impedance-matching effect in the 200–3000 nm band, thus exhibiting a broadband perfect absorption.

To thoroughly understand the absorption mechanism of the designed absorber, we selected five representative wavelengths, namely, 200, 500, 1000, 1800, and 2300 nm, and then investigated the electric field |E| and magnetic field |H| profiles in the x-z plane for y = 0 nm in Figure 4 and Figure 5, respectively. From Figure 4a, one can see that the local electric field is largely distributed on both sides of the top of the pyramidal structure at the wavelength of 200 nm. The electric field intensity located at the top of the pyramid structure gradually moves to the bottom, when the incident wavelength varies from 500 to 2300 nm (Figure 4b−e), and the electric field intensity gradually enhances with the increase of wavelength and fills the interspace of two nearby pyramid structures. Hence, the electric field is largely focused on the edge and the bottom corner of the Ti pyramid structure. The main reason for producing the localized strong field is the excitation of the localized surface plasmon resonances (LSPR) on the Ti pyramid [35,36,37]. Additionally, the electric field in the air gap between two adjacent unit cells is also significant due to the coupling between the two neighboring LSPR modes. Similarly, as can be observed from Figure 5, at a shorter wavelength, such as λ = 200 nm, the magnetic field is localized at the top part of the pyramid. As the incident wavelength increases, the distribution of magnetic field slowly moves down, which is similar to the change trend of the electric field. With the evolving widths from top to bottom, the pyramid can support resonances at different wavelengths, and the overlap of these resonances leads to a wideband enhanced resonant absorption. Consequently, in addition to the intrinsic loss of the Ti material, the excellent absorptivity of the designed absorber was derived from the LSPR effect of the Ti pyramid and the coupling of resonance modes between two nearby pyramids.

### 3.3. Tunable Absorption with Geometric Parameters

To gain a deeper understanding of the absorption characteristics of the designed absorber and to look for tolerance of manufacturing, we simulated the effect of the structure size on the absorption properties. As depicted in Figure 6, when the TE polarized light is vertically incident, the changes of different geometric parameter values (P, h_1_, h_2_, and h_3_) do not cause the change of absorbance in the UV to VIS wavelength range. 

Firstly, we investigated the effect of the period of unit cell P on the absorption spectrum of the designed structure. Except for the unit cell period, other geometric parameters are set as follows: h_1_ = 800 nm, h_2_ = 30 nm, and h_3_ = 110 nm, and the variation range of P is 100–180 nm. Figure 6a depicts the simulated absorbance as functions of the wavelength and P. By increasing the period of unit cell P from 100 to 180 nm, the absorption increases in the intermediate band of about 1238−1920 nm, decreases slowly in the short wavelength range of 500–1200 nm and decreases rapidly in the long wavelength range of 1921–3000 nm, with a blue shift. The primary cause of this phenomenon is the weakening of the LSPR at the base corner of the pyramid as well as the coupling between the two nearby resonant modes with the increase of the spacing between adjacent pyramids. Thus, the proposed absorber at P = 100 nm is one step closer to achieve broadband and perfect solar energy absorption. 

Subsequently, the height of the pyramid h_1_ also has a certain impact on the absorbing ability of the absorber. The absorption spectrum for the situation of an increasing height of the pyramid (h_1_) are shown in Figure 6b, where other parameters remain unchanged (P = 100 nm, h_2_ = 30 nm, and h_3_ = 110 nm). As the height of the pyramid is increased from 760 to 840 nm, slight spectral changes are observed in the short wavelength region (UV-VIS wavelength region), and long wavelength absorption changes become more obvious and undergo red-shift. In addition, the absorption decreases across the middle wavebands (1400–2000 nm) and increases across the short (1000–1400 nm) and long (2000–3000 nm) wavebands. Therefore, we take the middle height h_1_ = 800 nm as the optimum height of the pyramid.

As can be seen in Figure 6c, the absorption spectrum varies with the dielectric SiO_2_ layer thickness h_2_, while other parameters are fixed (P = 100 nm, h_1_ = 800 nm, and h_3_ = 110 nm). With the thickness of SiO_2_ layer increasing from 20 to 40 nm, the absorbance displays a monotonically increasing trend at 1600−2250 nm, while the absorbance exhibits a monotonically decreasing trend. Comparing Figure 6c,b, the absorption spectrum changes are completely opposite, and the former spectrum changes more drastically. The optimum layer thickness of dielectric SiO_2_ is determined to be 30 nm.

Because underlying metal is mainly used to block transmitted light, the substrate thickness is designed to be very thick in most studies. In order to further facilitate processing and save cost, we fixed other parameters (P = 100 nm, h_1_ = 800 nm, and h_2_ = 30 nm) for simulation to study the effect of the underlying metal thickness on absorption, as shown in Figure 6d. When h_3_ = 70 nm, the broadband low absorption region appears near 2600 nm, and the average absorption of the whole band is 99.71%. With the thickness of the Ti layer increasing from 90 to 200 nm, the absorption spectrum in the short wavelength region hardly changes, while the long wavelength spectrum changes more obviously with a red-shift. While bottom layer thickness h_3_ is 110 nm, the spectral curve is relatively flat, and the average absorption of 99.87% is slightly higher than 99.84% (h_3_ = 200 nm). The optimum value of the underlying metal thickness h_3_ is 110 nm to save cost.

In general, under a certain structural parameter tolerance, the designed absorber not only can maintain the broadband perfect absorption characteristics, but also is low-cost, easy to manufacture and beneficial to practical applications.

### 3.4. Polarization-Independent and Wide-Angle

It is significant that the polarization and the incident angle hold insensitivity to maximize absorption of solar energy for a perfect solar absorber in practical applications. We conducted a numerical simulation about the absorbability of the designed solar absorber in this paper with different polarization and incident angles, and the results are depicted in Figure 7a,b, respectively. Figure 7a shows that the absorption spectrum remains unchanged when the polarization angle changes from 0° to 90°, with an average absorption rate of 99.84%. This insensitivity of the solar absorber to the polarization is due to the high degree of the symmetry of structure. Figure 7b presents the absorption band with the incident light angle adopting the data range from 0° to 80° at an interval of 10°, and the absorber has absorptivity values of 99.84%, 99.86%, 99.50%, 97.80%, 94.33%, 88.85%, 84.24%, 82.08%, and 87.86%, respectively. In addition, the perfect wideband absorption spectrum almost shows stability absorption from 0° to 40°. For incident angles less than 40°, the absorption bandwidth remains essentially constant. Figure 7b presents the absorber exhibits stability absorption for incident angles less than 40° and the absorption bandwidth is unchanged essentially within 200−3000 nm. By further increasing the incidence angle to 50°, the absorption intensity begins to decrease slightly in the short wavelength band (near 800 nm). Nevertheless, when the incident angle is 80°, the absorber can still own an 87.86% absorption efficiency. At oblique incidence, as the angle of incidence increases, the scattering and the reflection increase, which in turn leads to a decrease in the absorption performance of the absorber, but the magnitude and the extent of the absorption intensity reduction are relatively limited. The above properties prove that the absorber is capable of broadband absorption when the angle of incidence varies greatly and provides polarization-independent absorption at the normal incidence of light. 

### 3.5. Influences of Different Materials on the Absorption Performance

Under the same geometrical structural parameters, with different metal and dielectric materials of the absorber were simulated. In addition, the absorption properties were compared with the designed absorber to evaluate the performance of the designed perfect absorber. For comparison, the absorption spectrum of the proposed (Ti) absorber presented in Figure 1b reappears in Figure 8 (red curve). Firstly, we researched the effect of metallic materials by replacing Ti metals with several common metals (Au, Ag, Cu, and Ge), as shown in Figure 8a. Here, the dielectric constants of Au, Ag, Cu, and Ge were taken from Refs. [32,38]. One can see from Figure 8a that the absorption properties of the Au, Ge, and Cu devices are near-unity in the UV to VIS band, whereas their absorption is much weaker than that of the Ti device in the NIR band. Additionally, the absorption of the Ag device is far weaker than that of the Ti device, with a more obvious difference, because it has the lowest material loss. Subsequently, other refractory metals were simulated to replace Ti, and the properties of the designed structures based on different refractory metal materials have been further investigated under the same size parameters. The permittivity values of zirconium (Zr), chromium (Cr), and iron (Fe) can be acquired from Refs. [31,32]. Figure 8b compares the absorption spectra of absorbers formed by various refractory metals (Zr, Cr, and Fe). One can see that the refractory metals Zr and Cr exhibit excellent absorption performance compatible to Ti. As shown in Figure 8(b), the absorption bandwidth of the Fe-based absorber is narrower compared to those of the Zr and Cr-based absorbers. Figure 8c shows the comparison between the designed Ti-based absorber and the other metal absorbers mentioned above in terms of the average absorption efficiency and broadband absorption of over 90%. The results show that our designed Ti-based absorber has the widest absorption bandwidth and its absorption rate is the closest to perfection.

Finally, we added aluminum sesquioxide (Al_2_O_3_), titanium dioxide (TiO_2_), and ferric oxide (Fe_2_O_3_) as further study objects. Al_2_O_3_, Fe_2_O_3_, TiO_2_, and SiO_2_ materials’ refractive indices were extracted from Palik’s database [32]. As seen in Figure 9, the minimal absorption even exceeds 90% from 200 to 3000 nm for all-dielectric spacers, suggesting an ultra-broadband near-perfect absorption window covering the main operation band of solar radiation. The structure has a high level of robustness against insulator materials. In addition, Al_2_O_3_, Fe_2_O_3_, and TiO_2_ are all high-temperature refractory materials, ensuring the stability of the absorber’s operation. The above results indicate that the designed metamaterial structure has a high degree of freedom in selecting diverse materials, and especially, the change of the insulation materials is quite stable.

## 4. Conclusions

In summary, we have designed and proved an omni-bearing, ultra-wideband-metamaterial perfect absorber based on periodic Ti pyramid arrays and a SiO_2_ dielectric film on a Ti substrate. The designed absorber exhibits a near-unity ultra-broadband absorption with an average absorbance of 99.84% and an absorption bandwidth over 2800 nm covering the band with the highest concentration of solar radiation energy. The efficient absorption is mainly derived from the intrinsic loss of the Ti material, the LSPR effect of the Ti pyramid, and the coupling of resonance modes between two nearby pyramids. Meanwhile, as the continuous width of the pyramid structure evolves from the apex to the base, LSPR modes of different wavelengths are generated, and the overlap of these modes result in an ultra-broadband absorption waveband. Additionally, the absorber exhibits high incidence angle tolerance and polarization insensitivity at normal incidence. In addition, the absorber exhibits a respectable level of robustness against various materials, which broadens the design and manufacturing options for the absorber. Our work provides a kind of ultra-wideband solar absorber which is very suitable for solar energy collection and capture and provides more possibilities for common applications of solar energy absorbers in practical production and life.

## Figures and Tables

**Figure 1 nanomaterials-13-00091-f001:**
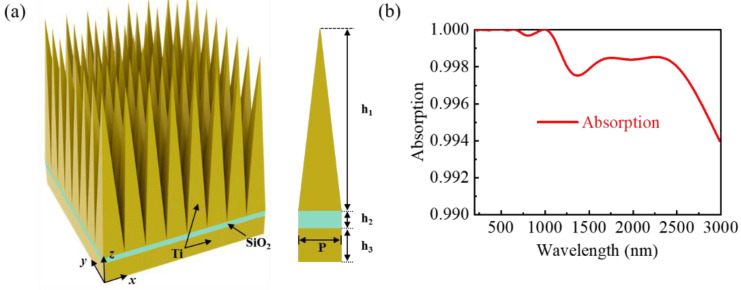
(**a**) Schematic diagram of the designed perfect absorber structure; (**b**) the corresponding absorption spectrum.

**Figure 2 nanomaterials-13-00091-f002:**
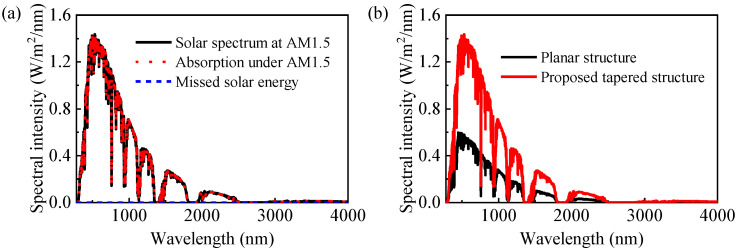
(**a**) Absorption spectra of the designed absorber in the solar spectrum at AM 1.5 illumination; (**b**) comparison of the absorption spectra of the designed structure and its planar counterpart in the solar spectrum at AM 1.5 illumination.

**Figure 3 nanomaterials-13-00091-f003:**
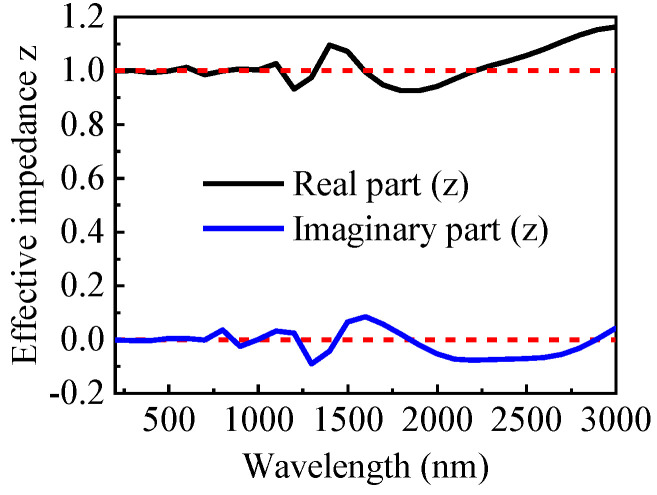
The effective impedance (Z) of the perfect absorber.

**Figure 4 nanomaterials-13-00091-f004:**
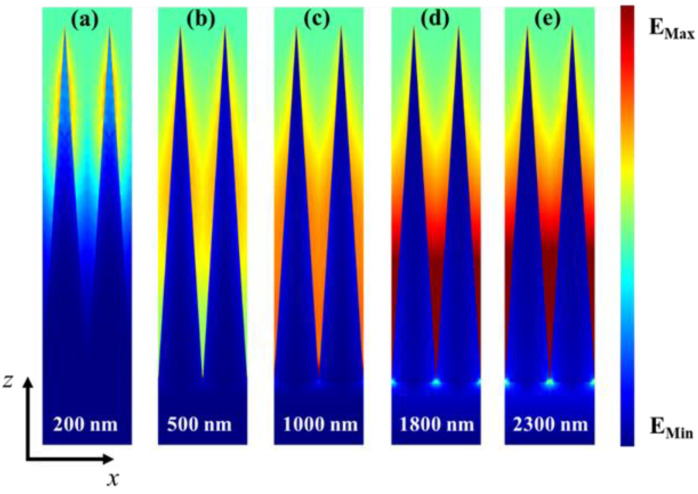
Electric field distributions at the different wavelengths of 200 nm (**a**), 500 nm (**b**), 1000 nm (**c**), 1800 nm (**d**), and 2300 nm (**e**) in the x-z plane.

**Figure 5 nanomaterials-13-00091-f005:**
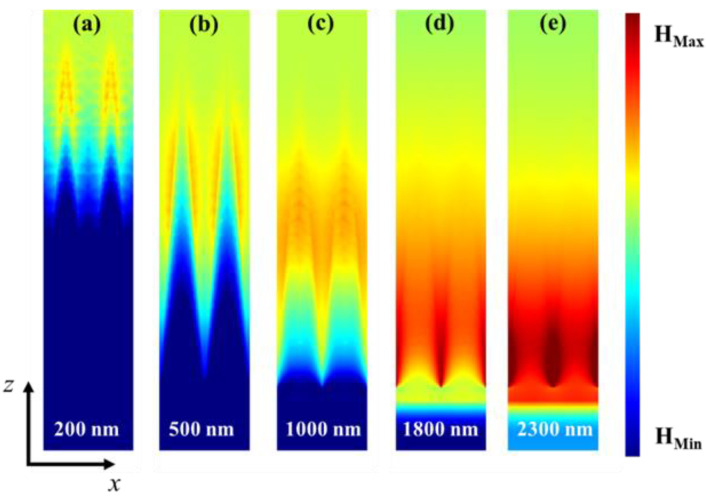
Magnetic field distributions at the different wavelengths of 200 nm (**a**), 500 nm (**b**), 1000 nm (**c**), 1800 nm (**d**), and 2300 nm (**e**) in the x-z plane.

**Figure 6 nanomaterials-13-00091-f006:**
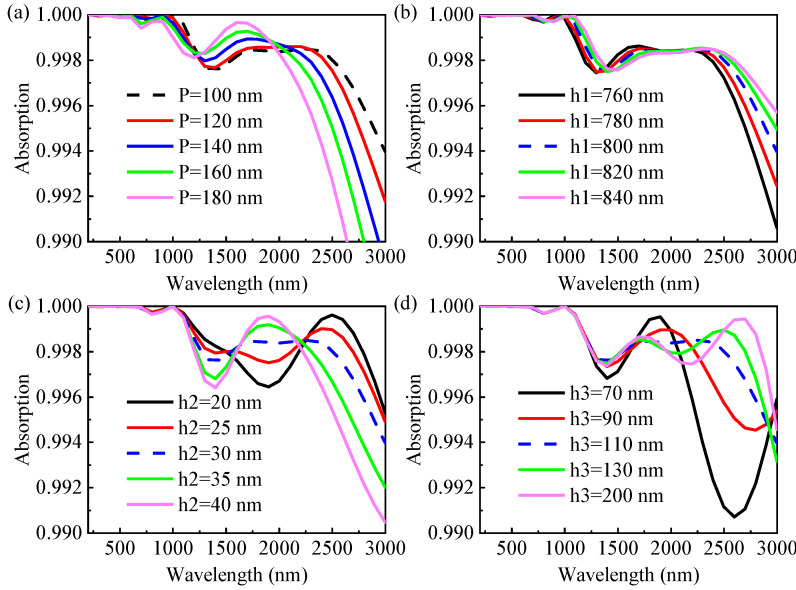
Effects of structural parameters on the absorbing properties: (**a**) period of unit cell P; (**b**) height of the pyramid h_1_; (**c**) thickness of SiO_2_ layer h_2_; and (**d**) thickness of the underlying metal Ti layer h_3_.

**Figure 7 nanomaterials-13-00091-f007:**
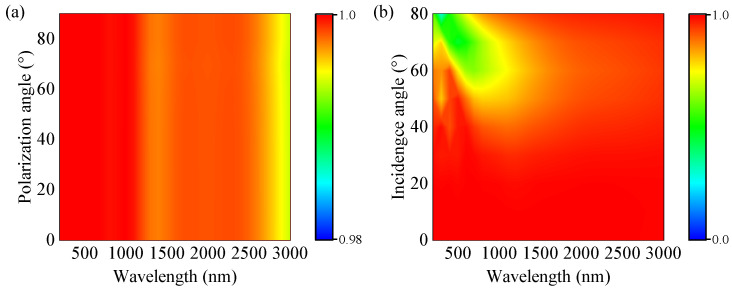
Absorption diagrams of the proposed solar energy absorber by tuning polarization (**a**) and incident angle (**b**).

**Figure 8 nanomaterials-13-00091-f008:**
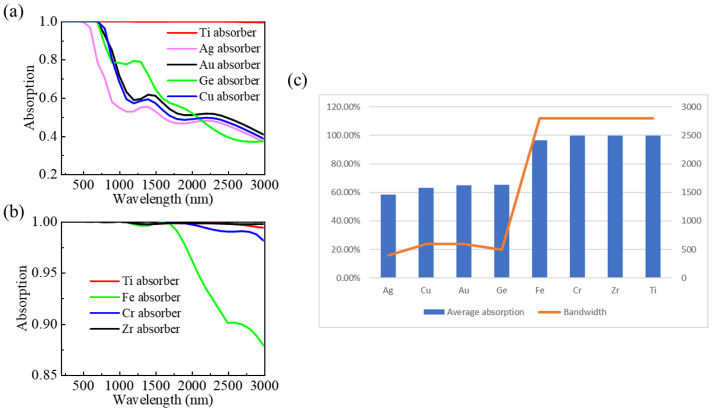
(**a**) Simulation results of different materials by keeping the SiO_2_ dielectric film unchanged and replacing Ti material with common metals Au, Ag, Cu, and Ge; (**b**) simulation results of different materials by keeping the SiO_2_ dielectric film unchanged and replacing the Ti material with refractory metals Zr, Cr, and Fe; (**c**) the average absorption efficiency and the broadband absorption of over 90%.

**Figure 9 nanomaterials-13-00091-f009:**
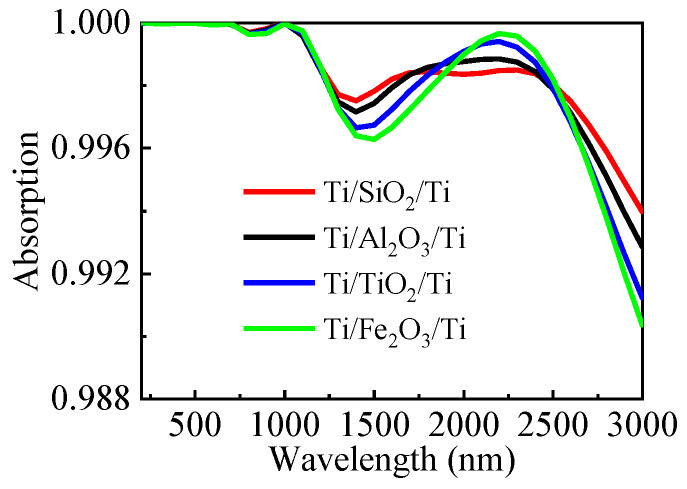
Simulation results of different materials by keeping metal Ti unchanged and replacing the SiO_2_ dielectric film with Al_2_O_3_, Fe_2_O_3_, and TiO_2_.

## Data Availability

Data underlying the results presented in this paper are not publicly available at this time but may be obtained from the authors upon reasonable request.

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
