# Peer review of "Ultra-Broadband Perfect Absorber based on Titanium Nanoarrays for Harvesting Solar Energy"

_nanomaterials, 2022, doi:10.3390/nano13010091_

Round 1
Reviewer 1 Report
Certain essential considerations are missing and must be addressed to be accepted for publication in this journal. Following are the comments and concerns:
1- In the introduction, you claim that your design will lower the fabrication cast compared to the other complex nanostructures. But it is obvious that the fabrication of the pyramid arrays will be costly according to the nm scale patterning. So, in your introduction and conclusion, you cannot use this sentence as the advantage of your design (Lines, 52-54 and 295). Besides, there are nanostructures with maximum absorption, which are so easier to fabricate.
2- In line 111 you used solar spectrum at AM 1.5. It is recommended to introduce AM 1.5 before starting to use the expression.
3- In Figures 4 and 5 you explained electric magnetic field distribution at different wavelengths. By increasing the wavelength, the distribution of electric fields and magnetic fields are enhanced and moved down from the tip to the bottom of the pyramids. But at the end (line 164) you did not explain properly how this will lead to excellent absorptivity in your structure where according to figure 1b the absorption of the structure is less in longer wavelength.
Author Response
Please check out word!

Reviewer 2 Report
The paper discusses the properties of an ultra-wideband perfect absorber for solar energy collection and capture. The absorber is based on the periodic Ti pyramid arrays and the SiO2 dielectric film on the Ti substrate. The absorber is able to operate from the ultraviolet to the near-infrared spectral range. It consists of a periodically aligned titanium nanoarray which sizes are selected by an optimal manner via detailed simulation. The average absorption efficiency of the optimal parameter absorber can reach up to 99.84% in the 200-3000 nm broadband range. This high efficiency, as the authors believe, comes mainly from the intrinsic loss of Ti material, when localized surface plasmon resonances of Ti pyramid realizes and also the coupling of resonance modes between two nearby pyramids. The absorber has high polarization independence, and large angular tolerance. The paper falls into the scope of the journal and contains new data. However, several improvements are necessary before publication.
- In the paper many times is written “metamaterial perfect absorbers” but the authors do not prove the metamaterial properties of the structure under study.
- The excitation of the localized surface plasmon resonances (LSPR) on the Ti pyramid is regarded as the main reason for the high absorbance efficiency. As a matter of fact, the presence of LSPR does not proved and the related references do not given.
- In order to calculate the absorbance of the structure, the dielectric constants of several metals including Ti are taken from the references. It seems much better if the corresponding dependences of the dielectric constants be given in the paper under review, and the comparison be outlined, showing the difference between Ti and the other metals and explaining the better absorbance with Ti pyramids.
- The confusing phrase in line 136 has to be corrected. The better absorption can be achieved when the value of the effective impedance amounts to 1.
- The Author Contributions should be corrected in the following part: “Yufang Liu … provide financial support”. Please, look the Instructions for Authors, see https://www.mdpi.com/journal/nanomaterials/instructions#authorship: “In order to qualify for authorship of a manuscript, the following criteria should be observed: Substantial contributions to the conception or design of the work; or the acquisition, analysis, or interpretation of data for the work”, etc.
- The misprint has to be corrected: Capture to Figure 7: “incident angel”.
Author Response
Please check out word!

Round 2
Reviewer 2 Report
The revised version of the paper is suitable to be published.